# Dose-Escalating (50–500 mg) Gluten Administration Leads to Detectable Gluten-Immunogenic-Peptides in Urine of Patients with Coeliac Disease Which Is Unrelated to Symptoms, a Placebo Controlled Trial

**DOI:** 10.3390/nu14091771

**Published:** 2022-04-23

**Authors:** Jordy P. W. Burger, Ellen G. van Lochem, Elisabeth A. Roovers, Joost P. H. Drenth, Peter J. Wahab

**Affiliations:** 1Department of Gastroenterology and Hepatology, Rijnstate Hospital, 6815 AD Arnhem, The Netherlands; pwahab@rijnstate.nl; 2Department of Microbiology and Immunology, Rijnstate Hospital, 6815 AD Arnhem, The Netherlands; evanlochem@rijnstate.nl; 3Department of Clinical Research, Rijnstate Hospital, 6815 AD Arnhem, The Netherlands; lroovers@rijnstate.nl; 4Department of Gastroenterology and Hepatology, RadboudUMC, 6525 GA Nijmegen, The Netherlands; joostphdrenth@cs.com

**Keywords:** coeliac disease, gluten immunogenic peptides, gluten excretion urine, gluten-free diet monitoring

## Abstract

Background: To determine the applicability and sensitivity of a urine self-test to detect gluten-immunogenic-peptides (GIP) in daily-life for patients with coeliac disease and correlate the test results with reported symptoms. Methods: We performed a prospective double-blinded placebo-controlled study, including adults with coeliac disease adhering to a strictly gluten-free diet. Patients were administered gluten in test-cycles of ascending doses of 50, 100, 200, and 500 mg alternated with placebo. Urine portions from 2, 5–17 h after the ingestion were collected and analyzed for GIP using the iVYCHECK-GIP-Urine rapid lateral flow test. Patients completed a diary mapping symptoms (nausea, bloating, diarrhea, abdominal pain, and lower level of energy). Results: We enrolled 15 patients and 7 received all 4 cycles with increasing gluten dosing. GIP was detected from urine in 47% of the patients receiving 50 mg gluten and in 86% with 500 mg gluten. We detected GIP in 20–50% of urine samples after placebo. There was no correlation between symptoms, gluten administration and/or GIP in urine. Conclusions: Gluten intake, even with a dose as low as 50 mg, leads to detectable urinary GIP concentrations. There is no correlation of coeliac disease ascribed symptoms with detection of urinary GIP.

## 1. Introduction

A gluten-free diet is the cornerstone of treatment of coeliac disease. It is associated with healing of mucosal disease, disappearance of serological antibodies (IgA class anti-tissue transglutaminase (TG2A), anti-endomysium antibodies (EMA)), and relief of symptoms [1]. Despite adherence to gluten-free diet, up to 20 percent of patients report persisting or recurrent symptoms such as abdominal pain, bloating, and diarrhea [2,3]. Gluten contamination in the diet may lead to overt symptoms; but there is great variability between patients in the degree of intolerance to gluten. A larger proportion of patients are convinced that their diet is strictly free of gluten and it remains uncertain whether their symptoms are due to ingestion of cryptogenic gluten contamination of food or are related to alternative sources of symptoms such as irritable bowel syndrome (IBS) [4].

The fact that symptoms may be related to (cryptogenic) gluten intake increases insecurity on patients’ part and results in avoidance of social activities such as holidays, dining out, and gatherings. The health-related quality of life (HRQoL) in patients with coeliac disease is lower when persistent symptoms are present in combination with self-reported problems with dietary adherence [2].

As anti-TG2A and -EMA are unreliable for incidental exposure of small amounts of gluten [5], these tests are not useful to check for inadvertent gluten intake. Recently the iVYCHECK GIP Urine rapid lateral flow test and iVYCHECK GIP stool kit (Biomedal, Seville, Spain) became available as a self-test to detect gluten immunogenic peptide (GIP) in urine or stool [6,7]. Detecting GIP in urine may provide real-time monitoring of exposure and aid patients in deciding whether symptoms may be attributed to inadvertent gluten exposure [8,9]. This will inform patients on possible dietary mistakes or inadvertent gluten intake, and afford better control in self-management of the disease.

Therefore, we aim to determine the applicability and diagnostic properties of this GIP urine self-test in daily life for patients with coeliac disease in relation to self-reported symptoms.

## 2. Materials and Methods

### 2.1. Study Population

This prospective double-blinded placebo-controlled cross-over study, included adult patients with coeliac disease adherent to a gluten-free diet for at least one year prior to inclusion. Patients were recruited at the outpatient clinic of the Rijnstate Hospital, Arnhem, The Netherlands. As per investigational review board instructions, no more than 20 patients were allowed to enter this trial, in order to limit the cohort that would be intentionally exposed to gluten.

All patients were diagnosed with positive serology (i.e., IgA class-tissue transglutaminase (TG2A) antibodies and anti-endomysium antibodies (EMA)) and/or biopsy-proven coeliac disease, histologically Marsh 2 or higher. Patients with Marsh 2 enteropathy without positive serology were only included when HLA DQ2/DQ8 genotype was supportive of coeliac disease and patients had a positive clinical response to the gluten-free diet. If patients were diagnosed at childhood age, the ESPGHAN diagnostic-criteria were followed. According to the Oslo revision of the ESPGHAN guidelines (2012), diagnosis in children may be made on serology, HLA genotyping, and clinical response to a gluten-free diet, without histology [10]. All patients reported strict dietary adherence, without ingestion of gluten-containing product and described prolonged episodes without symptoms related to coeliac disease alternated with periods of abdominal pain, bloating, diarrhea, and/or lower level of energy after suspected gluten contamination. Standard of care included a referral to a dietitian with expertise in the field for nutritional assessment, diet education, meal planning, assistance with the adaptation to the challenging new gluten-free lifestyle, and regular dietician guided dietary review. Patients had access to the Dutch coeliac disease patient associations’ resources on gluten-free diet. Patients were encouraged to participate in peer support groups dedicated to patients management of coeliac disease. TG2A and EMA antibodies assessed at the last follow-up visit less than 3 months prior to inclusion were undetectable. No GIP were detected in a spot urine using the iVYCHECK GIP Urine rapid lateral flow test (Biomedal, Seville, Spain) at baseline. We excluded patients diagnosed with refractory coeliac disease or any alternative gastroenterological disease that could explain abdominal symptoms. Furthermore, patients who were pregnant, used painkillers, or other medication to reduce gastrointestinal symptoms were excluded. The study protocol was reviewed and approved by the investigational review board Arnhem Nijmegen and registered on clinicalTrials.gov (NL2016-2877).

### 2.2. Test Period

All patients provided a baseline urine sample on the day they provided informed consent. At the start of the test period, patients were handed a set of three identical capsules (Vegetarian HPMC size 1 white; (filled by Microz BV, Geleen, The Netherlands) containing either a total amount of 50 mg of gluten supplemented with cellulose or placebo (cellulose) (batch manufacturing and testing records and supplier’s certificate of analysis are available upon request). Both patient and researcher were blinded for the content of the capsules. Patients were asked to ingest the capsules with water at t = 0, just after waking up in the morning. Symptoms were registered in a diary from the day prior to ingestion up to 48 h after ingestion of the capsules (see Figure 1). Urine produced between 2, 5, and 17 h after capsule ingestion was collected.

One week after the first set of capsules, the alternating set of capsules were provided containing either 50 mg of gluten (after a first set with placebo) or placebo (after a first set with gluten). The procedure for patients was identical to the first week. After this second ingestion, a two-week wash-out period followed. Because earlier research found that higher doses of gluten afforded better detection of GIP in urine of healthy individuals and the amount of gluten contamination causing symptoms in patients was unknown, we chose to test at increasing doses. An identical cycle was repeated with capsules containing (100 mg, 200 mg, and 500 mg) of gluten or placebo for a maximum of four test-cycles (i.e., placebo and gluten) during a 12-week period (see Figure 1). Upon request of the investigational review board, a patient could leave the study if symptoms were reported during two consecutive cycles (regardless of gluten and placebo). Patients were asked to inform the research team whenever they thought to have been accidently exposed to gluten during the study period.

### 2.3. Urine Sample Collection

All urine samples were stored in the refrigerator at the residence of the patient until central collection that followed within 24 h. During transportation, that took less than 2 h, samples were stored in a cooling box ensuring temperature would not exceed 8 °C. After arrival at the laboratory, the samples were stored in a refrigerator at a temperature of 4 to 6 °C. Within 3 days after arrival, the samples were deep-frozen and stored at a temperature of −20 °C until further processing.

### 2.4. Urine Sample Analyses

We measured GIP in urine using the iVYCHECK GIP Urine rapid lateral flow test (Biomedal, Seville, Spain). This specific immunochromatographic assay has been developed for home use. Samples were processed and tested according to the manual and manufacturers’ instructions. Briefly, a small amount of urine is mixed with a conditioning solution and exposed to a cellulose strip where the gluten peptides in the samples react with monoclonal anti-gliadin 33-mer red-colored microsphere. After spreading by capillarity, the peptide–conjugate complex reaches the test zone where it interacts with a second anti-gliadin 33-mer antibody immobilized on the membrane. This results in a red line when the urine contains GIP. A green line is visible as a control of the process. A special calibrated reader was used to quantitate GIP concentrations in urine. The detection limit of the reader is 2 ng/mL.

### 2.5. Symptom Diary and Questionnaires

We registered patients self-reported dietary adherence. During the test-cycles, patients filled in a diary regarding their symptoms over the last 24 h (addressing nausea, bloating, diarrhea, abdominal pain, and lower level of energy on a 5-point Likert-scale. The scale ranged from none of the time (0 points), a little of the time (1 point), some of the time (2 points), most of the time (3 points) and all of the time (4 points)). Scores were collected from the day before up and until 48 h after ingestion of the capsules. The specific questions and answer options were taken from the celiac symptom index [11], in the USA validated questionnaire regarding symptoms over the last 4 weeks and adapted to a 24-h setting.

### 2.6. Statistical Analysis and Data Management

We present descriptive statistics as mean with standard deviation for normally distributed continuous data, median and inter-quartile range for skewed continuous variables, and as numbers and percentages for dichotomous and categorical variables. Visualization of the output includes boxplots and radar charts. A radar chart is a graphical method used for displaying multivariate data, represented on axes starting from the same point. Patients were randomized using a web-based randomization service (Sealed Envelope). Available online: https://www.sealedenvelope.com/simple-randomiser/v1/lists (accessed on 19 September 2019). Statistical analyses were performed with IBM SPSS statistics 25.0. Data are stored on the Rijnstate Hospital server. Data captured on paper, have been stored in a key-locked closet. We transferred data originally entered on paper into Research Manager (4.0–6.0), which we used for data management and data monitoring purposes. Auditing was performed on structural basis following the principles of good clinical practice. We warrant privacy by using encrypted and unique individual codes which are stored separately from the study data. Data will be saved for 15 years after termination of the study (2021).

## 3. Results

During the study period 18 patients were eligible for inclusion in the study. Two patients did not start the study, one patient moved to another city, and another patient declined because it was difficult to commit time to the project. Furthermore, one patient was excluded for reasons that the patient did not adhere to the protocol. Finally, 15 patients completed at least one test round, 7 patients all four cycles. See Figure 2 for flow of patients through the study. Patient characteristics are given in Table 1.

### 3.1. Detection of GIP in Urine

We analyzed a total of 326 urine samples from 15 different patients who participated. All patients were tested at baseline with no detectable GIP in urine. The median number of urine samples per patient was 4 (range 1–6). Patients who collected more than one urine sample during the 14.5 h sampling time were more likely to have at least one tested positive for GIP.

In 7 out of 15 patients (47%), at least one urine sample contained GIP > 2 ng/mL after ingestion of 50 mg of gluten. However, GIP was also detected in the urine of 3 out of 15 patients (20%) that had ingested the matching placebo. After administration of 100 mg of gluten, 8 out of 14 patients (57%) had detectable urinary GIP compared to 7 out of 14 patients (50%) after 100 mg of placebo. The 200 mg gluten ingestion led to detectable GIP in the urine of 5 out of 9 patients (56%) compared to 3 out of 9 patients (33%) after 200 mg of placebo. Finally, after administration of 500 mg of gluten, we detected urinary GIP in 6 out of 7 patients (86%) compared to 3 out of 7 (43%) after 500 mg of placebo. The results are summarized in Table 2.

### 3.2. Optimal Time of Urinary Sample

GIP was detectable in urine samples that were collected from 152–1020 min after ingestion of gluten. Highest concentrations were obtained from samples collected between 270–980 min after ingestion, see Figure 3.

### 3.3. GIP Concentration in Urine Compared to Gluten Dose

The median concentration of GIP in urine ranged from 0 (range 0–35.34) ng/mL after 50 mg of gluten to 11.39 (range 0–15.29) ng/mL after 500 mg of gluten using the urine samples with the highest concentration in that particular cycle. After receiving 50 mg, 200 mg, and 500 mg of matching placebo capsule, median GIP in urine was 0 (range 0–25.22) ng/mL. After receiving 100 mg matching placebo, median urinary GIP concentration was 3.01 (range 0–12.02 ng/mL), See Figure 4.

### 3.4. Symptoms before and after Gluten Administration

Symptoms were recorded 24 h prior to (day −1), the first 24 h after (day 0) and following 24 to 48 h after administration (day 1) of capsules in all patients receiving gluten and in patients without detectable GIP in urine after placebo. During the cycles of 50, 100, and 200 mg, all recorded symptoms were in the range between little of the time and some of the time. There was no evident correlation between symptoms and dose of gluten. After 500 mg of gluten, abdominal pain was reported most of the time during the first 24 h after ingestion in a sample of six patients. Abdominal pain was more prominent 24–48 h after ingestion of 100 and 200 mg of placebo with negative GIP in urine. Analysis of individual data does not give a more specific direction upon the relation between gluten dose, GIP, and symptoms.

The data of the test cycles for 50, 100, 200 and 500 mg are summarized in Figure 5.

### 3.5. Symptoms and Detectable GIP in Urine

Bloating and low level of energy are reported most of the time for the first 24 h after ingestion of 50 mg of gluten in the cohort of seven patients receiving gluten and a subsequent urinary GIP. After 100 mg of gluten and a subsequent urinary GIP in the cohort of eight patients, bloating is reported some of the time. About 24 to 48 h after 100 mg of placebo and negative GIP, low energy and abdominal pain are reported some of the time in a cohort of seven. After 500 mg of gluten, abdominal pain was reported most of the time during the first 24 h after ingestion in a sample of five patients. In-depth analysis found no consistent increase in scores related to gluten exposure at the individual patient level. 

The data of the test cycles for 50, 100, 200, and 500 mg are summarized in Figure 6.

## 4. Discussion

In this paper, we describe a placebo-controlled pilot study to assess applicability and diagnostic capabilities of the iVYCHECK GIP Urine rapid lateral flow test in coeliac patients in a real life setting. We studied increasing doses of gluten from 50 to 500 mg, mimicking inadvertent gluten intake in relation to reported symptoms.

Urine GIP was detected in 47% of patients after ingestion of 50 mg of gluten and in 86% after ingestion of 500 mg. These results are in line with earlier studies conducted in healthy individuals who reported significantly higher rates of GIP excretion in urine after higher doses of gluten [6]. The number of patients with detectable GIP in urine however is higher compared to studies addressing GIP excretion with lateral flow tests in urine and feces in healthy individuals, which found GIP in 15 and 22 percent after 50 mg of gluten [6,7].

All patients enrolled in this pilot study had negative urine GIP and negative TG2 A and EMA, at baseline and all were experienced with the gluten-free diet and clearly instructed to maintain restrictions during the study period. Nevertheless, GIP was discovered from urine samples in up to 20–50% of patients receiving placebo. These results are in line with data from a cross-sectional study describing detection of GIP in feces in a large group of patients with coeliac disease adhering to a gluten-free diet and suggest that gluten contamination with food ingestions is a reality in daily life [12]. Inadvertent gluten intake, even in small doses may account for a higher proportion of positive tests particularly at lower doses of gluten. This is supported by a recent study which found that healthy adults adhering to a normal gluten-free diet allowing “gluten-free” products containing <20 parts per million (PPM) led to more positive GIP in urine compared to a zero gluten diet [13].

We found that GIP was present in urine samples between 2, 5, and 17 h after ingestion of gluten. The majority of positive urine samples were collected between 3 and 12 h after ingestion and all patients who had at least one positive urine test result after gluten exposure collected urine in this time frame.

We did not see any consistent pattern in symptom reporting in patients who received gluten in any dose and/or had positive GIP in urine compared with their baseline symptoms. Symptoms were reported before and after intake regardless of placebo, gluten dosage, or presence of GIP in urine. Patients with coeliac disease responded with nausea and vomiting within 4 h after ingestion of 5 g of gluten [14]. Others describe symptoms such as nausea, vomiting, bloating, and diarrhea after gluten challenge with doses of 3 and 10 g of gluten daily [15]. Clearly, large doses of gluten are able to elicit symptoms of intolerance. In our study, we aimed to investigate smaller amounts of gluten intake, reflecting unexpected gluten contamination in daily life. Apparently, cryptogenic doses of gluten may be present even in “strict” gluten-free diet and the mere presence of gluten may trigger symptoms. Gluten-related symptoms overlap greatly with that related to irritable bowel syndrome which may be provoked by gluten or follow a totally different pathway and time frame.

A major strength of our study is that it addresses the correlation between proven gluten ingestion with control of urine samples for detectable GIP and coeliac-related symptoms in patients adhering to a gluten-free diet. We followed a placebo controlled double blinded design to mimic a daily life setting with regard to accidental gluten exposure. All urine results were quantified with a reader in the hospital and a portion of the urines were double checked. Finally, the results of our study bear clinical relevance since up to 20 percent of patients with coeliac disease continue to report intermittent symptoms that lowers their HRQoL. More insight into the cause of these symptoms may give them more control over their lives.

This study comes with limitations. First, in our study, in addition to the restricted amounts of gluten or placebo, we had no control over dietary intake although we instructed patients to avoid food items with a high risk of gluten contamination. Although all patients were educated on gluten-free diet by an expert dietician and were given access to resources to support a gluten-free diet, trace amounts of GIP was detectable from patients after intake of the placebo capsules. Gluten contamination interferes with the study outcome, but this mirrors exactly the real-life situation that led to the conception of this study. A recent study detected positive GIP in urine of healthy individuals adhering to a gluten-free diet but were allowed to eat gluten-free products containing <20 PPM gluten. This suggests that GIP testing is very sensitive in some cases and suggests that a positive GIP in urine does not always imply dietary mistakes considering the diet we prescribe [13]. On request of the investigational review board, we included a small sample of participants which limits the power of the study. We did not monitor fluid intake and it is possible that variations in fluid intake affect GIP concentration in urine resulting in false negative tests. While the urine samples were subjected to a number of freeze thaw cycles, this does not affect the test properties as GIP remains stable under these conditions. While our study explores the diagnostic merits of urinary GIP testing, a number of issues still need to be resolved. It is unclear which biomaterial (feces or urine) is best to detect gluten consumption. In addition, the latency between gluten exposure and GIP excretion is unknown.

## 5. Conclusions

In this pilot study with real life patients on a strict gluten-free diet, who were tested with increasing amounts of either gluten or placebo, the GIP urine test was not able to discriminate between urine GIP values due to gluten challenge or background contamination nor between symptoms related to direct gluten intake, prolonged gluten intolerance, or irritable bowel syndrome. Gluten intake, even with a dose as low as 50 mg, leads to detectable urinary GIP concentrations. Our data do not support the use of a GIP spot urine self-test to distinguish between gluten-related symptoms or other causes of symptoms in patients with coeliac disease.

## Figures and Tables

**Figure 1 nutrients-14-01771-f001:**
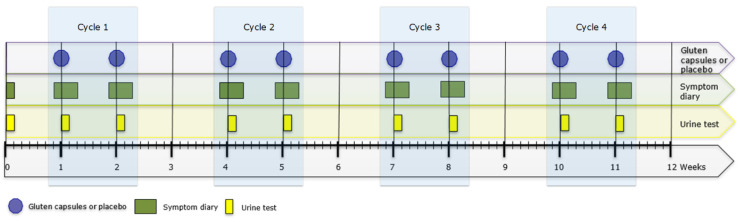
Study protocol flowchart.

**Figure 2 nutrients-14-01771-f002:**
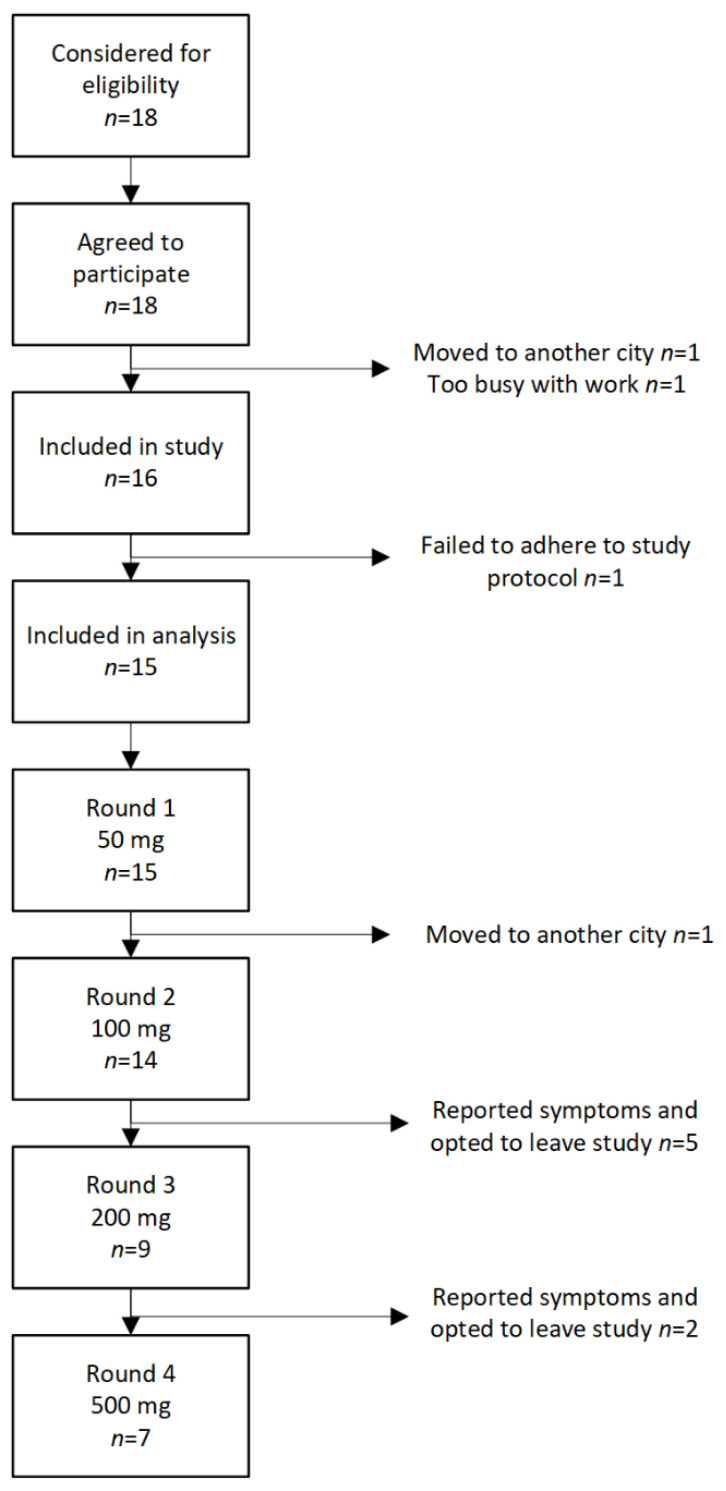
Flowchart of patients through the study.

**Figure 3 nutrients-14-01771-f003:**
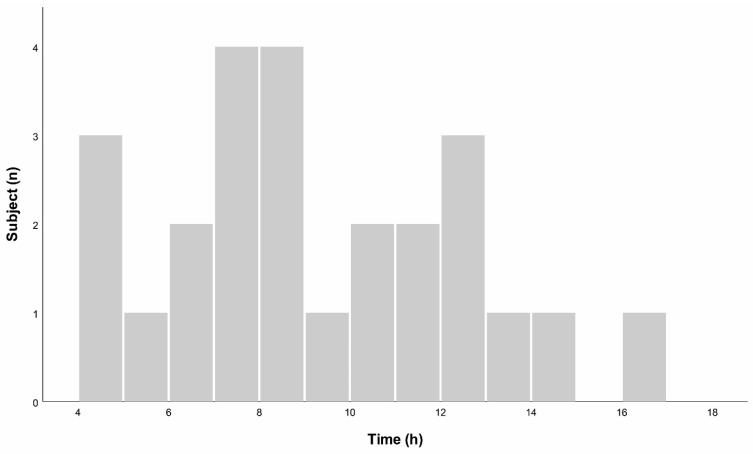
Time after gluten ingestion plotted against peak urinary gluten immunogenic peptides (GIP) concentration. *x*-axis: time in hours. *y*-axis: number of subjects.

**Figure 4 nutrients-14-01771-f004:**
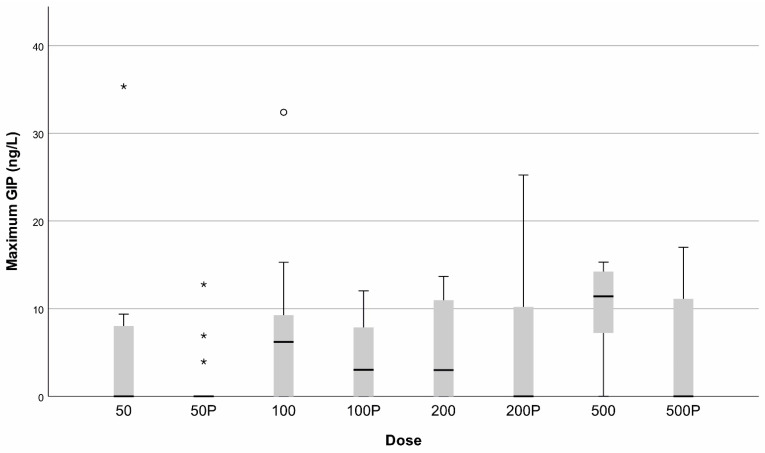
Maximum concentration of gluten immunogenic peptides (GIP) in ng/mL in urine after gluten or placebo ingestion. *x*-axis: dose of gluten or placebo (p). *y*-axis: concentration of GIP in urine in ng/L. Values more than three interquartile range from the end of a box are labeled as extreme, denoted with an asterisk (*). Values more than 1.5 but less than 3 interquartile range from the end of the box are labeled as outliers (o).

**Figure 5 nutrients-14-01771-f005:**
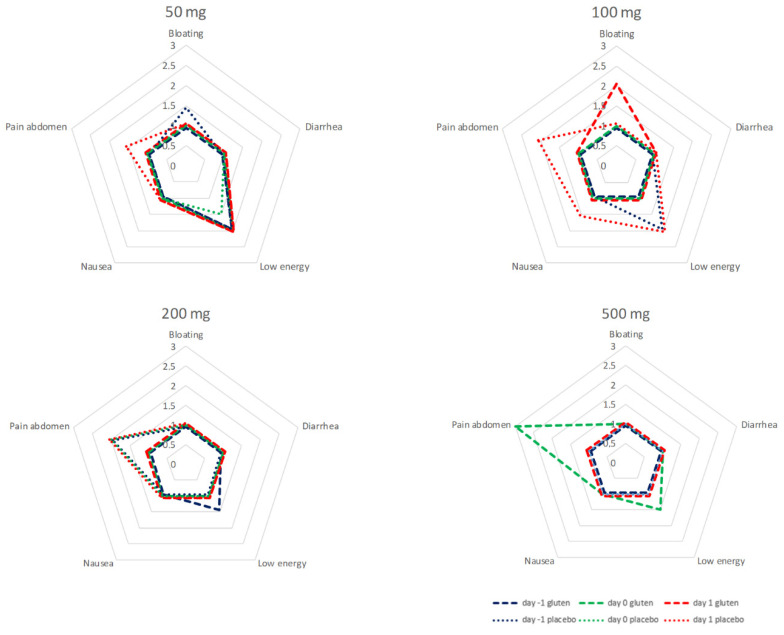
Symptoms from 24 h before until 48 h after ingestion of 50 mg of gluten and placebo. All patients that had received gluten were included irrespective of GIP in urine. Only patients that had negative urine for GIP after placebo were included. *x*-axis: the median answer on the 5-point Likert-scale (ranging from none of the time (0 points), a little of the time (1 point), some of the time (2 points), most of the time (3 points) and all of the time (4 points)) per cycle.

**Figure 6 nutrients-14-01771-f006:**
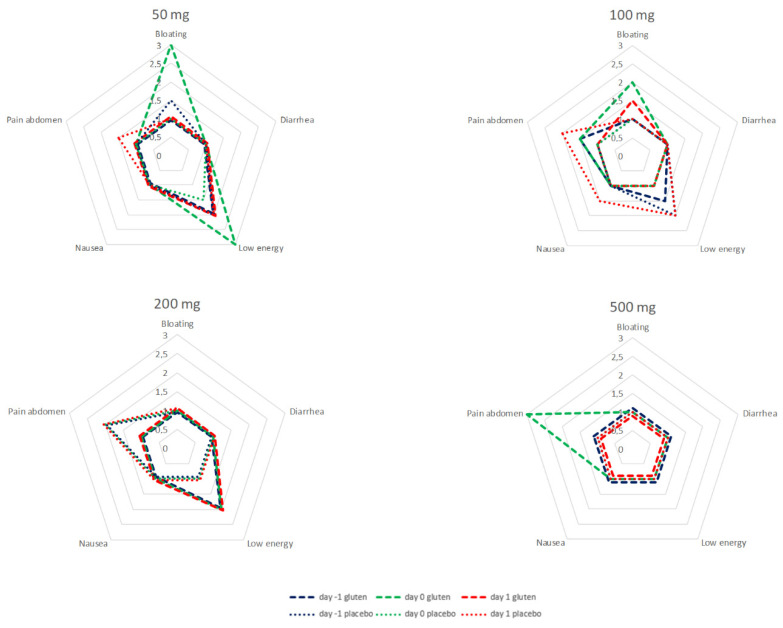
Symptoms from 24 h before until 48 h after ingestion of 50, 100, 200, and 500 mg of gluten and placebo. Only patients with positive GIP after gluten and patients with negative GIP after placebo were included in the analysis. *x*-axis: the median answer on the 5-point Likert-scale (ranging from none of the time (0 points), a little of the time (1 point), some of the time (2 points), most of the time (3 points) and all of the time (4 points)) per cycle.

**Table 1 nutrients-14-01771-t001:** Patient characteristics.

	Patients *n* = 15
Age (years)	44.7 (22–75)
Sex (Male/Female)	7 (47%)/8 (53%)
Marsh	
II	0
III A or higher	15 (100%)
no biopsy	0
Gluten free diet adherence (years)	7.2 (2–21)
CD ^1^ specific antibodies at baseline	
EMA ^2^ and/or TTG ^3^ positive	0 (0%)
EMA and/or TTG negative	15 (100%)
GIP ^4^ detectable in urine at baseline	
-Yes	0 (0.0%)
-No	15 (100%)

^1^ Coeliac disease, ^2^ endomysial autoantibodies (no normal values available) ^3^ tissue transglutaminase [<7 mmol/L], ^4^ gluten immunogenic peptide.

**Table 2 nutrients-14-01771-t002:** Gluten immunogenic peptide (GIP) excretion in urine samples.

	1	2	3	4	5	6	7	8	9	10	11	12	13	14	15		
50 mg ^1^	+	+	−	−	+	+	−	−	+	−	−	+	+	−	−	7/15	0.47
50 p ^2^	−	−	−	−	−	−	−	−	−	−	−	+	−	+	+	3/15	0.2
100 mg	+	+	+	−	−	+	−	−		−	+	+	+	−	+	8/14	0.57
100 p	−	+	−	+	+	−	+	−		−	−	+	+	−	+	7/14	0.50
200 mg	+	+	+	−	−	+		−			−			+		5/9	0.56
200 p	−	−	+	−	+	−		+			−			−		3/9	0.33
500 mg	+			+	+	+		+			+			−		6/7	0.86
500 p	−			+	−	+		+			−			−		3/7	0.43

Legend: ^1^ dosage of gluten containing capsules administered. ^2^ dosage of cellulose (placebo). + GIP detected in one spot urine sample. − no GIP detected in urine.

## Data Availability

Data available on request due to restrictions e.g., privacy or ethical. The data presented in this study are available on request from the corresponding author. The data are not publicly available.

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
