# Peer review of "Dose-Escalating (50–500 mg) Gluten Administration Leads to Detectable Gluten-Immunogenic-Peptides in Urine of Patients with Coeliac Disease Which Is Unrelated to Symptoms, a Placebo Controlled Trial"

_nutrients, 2022, doi:10.3390/nu14091771_

Round 1
Reviewer 1 Report
This is an interesting study on the clinical performance of GIP urine test in celiac patients in a real life setting. Measurement of GIP in urine or stool has been introduced as a tool to detect recent inadvertent gluten ingestion in patients adhering to a gluten-free diet. GIP are available for use in clinical settings and for disease self-managing by the patient, but recent studies confirm that some questions remain to be answered: the best sample to use (urine or stool?), the latency between gluten exposure and excretion in stool/urine, the relationship between the quantity of ingested vs eliminated gluten in stool/urine. In this study, the authors found no correlation between symptoms, gluten administration and/or GIP in urine, while, of note, GIP were detected also in 20-50% of urine samples after placebo. It is a hot topic in current celiac research and within the scope of the journal. The manuscript is clearly written, and the authors gave detailed description of the obtained results.
Major revision
One suggestion is revision of the References, adding the most recent literature on this topic for a comprehensive discussion. In particular, there are two studies in which subjects were administered multiple gluten challenges before testing GIP:
Coto L, et al. Individual variability in patterns and dynamics of fecal gluten immunogenic peptides excretion after low gluten intake. Eur J Nutr. 2022 Jan 7:1–17. doi: 10.1007/s00394-021-02765-z.
Monachesi C, et al. Determination of Urinary Gluten Immunogenic Peptides to Assess Adherence to the Gluten-Free Diet: A Randomized, Double-Blind, Controlled Study. Clin Transl Gastroenterol. 2021 Oct 6;12(10):e00411. doi: 10.14309/ctg.0000000000000411.
Minor Revision
I would suggest to add the abbreviation for “coeliac disease” (CD).
Discussion
Line 384: the authors cite reference [13] but it is not reported in the References list.
Author Response
Dear reviewer. We are grateful for the time and energy you have invested in our manuscript which resulted in valuable feedback. Your suggestions have improved our paper significantly. We have revised the manuscript and tried to improve the english.
C1 “One suggestion is revision of the References, adding the most recent literature on this topic for a comprehensive discussion. In particular, there are two studies in which subjects were administered multiple gluten challenges before testing GIP”
R1 We thank you for this suggestion. We added the suggested to the manuscript in both the introduction and discussion where we’ve placed our results in perspective to the most recent literature. See the highlighted lines 52-54, 313-316 and 325-328 in the revised manuscript
C2 “Line 384: the authors cite reference [13] but it is not reported in the References list.”
R2 We thank you for this comment. The missing reference, now number 15 is re-added to the reference list.
C3 "I would suggest to add the abbreviation for “coeliac disease” (CD)"
R3 We thank you for this comment. We will consider this in the uploaded manuscript
Reviewer 2 Report
The study is well constructed and well argued. The interpretation of the data is correct and appropriate and concludes by making a good self-criticism on the limits of the study. What is missing is the control of an expert nutritionist regarding the correct adherence to the diet and also the control over the quality and safety of the commercial gluten-free foods used.
The limitations of the ethics committee have limited the execution of a larger work that would have been necessary to have more comprehensive answers.
Author Response
We are grateful for the time and energy you have invested in our manuscript which resulted in valuable feedback. Your suggestions have improved our paper significantly. You will find the reponse to your comment below.
C1 “What is missing is the control of an expert nutritionist regarding the correct adherence to the diet and also the control over the quality and safety of the commercial gluten-free foods used”
R1 We thank you for this comment and understand this could be a pitfall. However, standard of care for the subjects involved in this study included a referral to a dietitian with expertise in the field for nutritional assessment, diet education, meal planning, assistance with the adaptation to the challenging new gluten-free lifestyle, and regular dietician guided dietary review. Patients had access to the Dutch coeliac disease patient associations’ resources on gluten-free diet. Patients were encouraged to participate in peer support groups dedicated to patients management of coeliac disease. We’ve tried to further clarify this in the paper see lines 81-87 and 358-366